# Tuberculosis Epidemiology and Badger (*Meles meles*) Spatial Ecology in a Hot-Spot Area in Atlantic Spain

**DOI:** 10.3390/pathogens8040292

**Published:** 2019-12-10

**Authors:** Pelayo Acevedo, Miguel Prieto, Pablo Quirós, Isabel Merediz, Lucía de Juan, José Antonio Infantes-Lorenzo, Roxana Triguero-Ocaña, Ana Balseiro

**Affiliations:** 1Instituto de Investigación en Recursos Cinegéticos (IREC) UCLM-CSIC-JCCM, 13071 Ciudad Real, Spain; pelayo.acevedo@uclm.es (P.A.); Roxana.triguero@uclm.es (R.T.-O.); 2Servicio Regional de Investigación y Desarrollo Agroalimentario (SERIDA), Centro de Biotecnología Animal, 33394 Deva-Gijón, Asturias, Spain; jmprieto@serida.org; 3Consejería de Infraestructuras, Ordenación del Territorio y Medio Ambiente, 33005 Oviedo, Asturias, Spain; pablo.gonzalez-quirosmenendezdeluarca@asturias.org; 4Laboratorio Regional de Sanidad Animal del Principado de Asturias, 33201 Gijón, Asturias, Spain; Isabel.meredizgutierrez@asturias.org; 5Centro de Vigilancia Sanitaria Veterinaria (VISAVET) y Departamento de Sanidad Animal. Universidad Complutense de Madrid, 28040 Madrid, Spain; dejuan@visavet.ucm.es; 6Unidad de Inmunología Microbiana, Centro Nacional de Microbiología, Instituto de Salud Carlos III, 28029 Madrid, Spain; 7Departamento de Sanidad Animal, Universidad de León, 24071 León, Spain

**Keywords:** badger, *Meles meles*, cattle, tuberculosis, epidemiology, ecology, multi-host system

## Abstract

We provide a temporal overview (from 2012 to 2018) of the outcomes of tuberculosis (TB) in the cattle and badger populations in a hot-spot in Asturias (Atlantic Spain). We also study the badger’s spatial ecology from an epidemiological perspective in order to describe hazardous behavior in relation to TB transmission between cattle and badgers. Culture and single intradermal tuberculin test (SITT) were available for cattle as part of the National Program for the Eradication of TB. A field survey was also carried out in order to determine the paddocks and buildings used by each farm, and the information obtained was stored by using geographic information systems. Moreover, eighty-three badgers were submitted for necropsy and subsequent bacteriological studies. Ten badgers were also tracked, using global positioning system (GPS) collars. The prevalence of TB in cattle herds in the hot-spot increased from 2.2% in 2012 to 20% in 2016; it then declined to 0.0% in 2018. In contrast, the TB prevalence in badgers increased notably (from 5.55% in 2012–2015 to 10.64% in 2016–2018). Both cattle and badgers shared the same strain of *Mycobacterium bovis*. The collared badgers preferred paddocks used by TB-positive herds in spring and summer (when they were more active). The males occupied larger home ranges than the females (Khr95: males 149.78 ± 25.84 ha and females 73.37 ± 22.91 ha; Kcr50: males 29.83 ± 5.69 ha and females 13.59 ± 5.00 ha), and the home ranges were smaller in autumn and winter than in summer. The averages of the index of daily and maximum distances traveled by badgers were 1.88 ± (SD) 1.20 km and 1.99 ± 0.71 km, respectively. One of them presented a dispersive behavior with a maximum range of 18.3 km. The most preferred habitat was apple orchards in all seasons, with the exception of winter, in which they preferred pastures. Land uses and landscape structure, which have been linked with certain livestock-management practices, provide a scenario of great potential for badger–cattle interactions, thus enhancing the importance of the badgers’ ecology, which could potentially transmit TB back to cattle in the future.

## 1. Introduction

Animal tuberculosis (TB) is a multispecies chronic infectious disease caused by members of the *Mycobacterium tuberculosis* complex (MTC), and principally by *Mycobacterium bovis* and *M. caprae*. Many countries have attempted to eradicate TB, thus allowing the European Commission to declare some of them officially free [1]. However, despite extensive control measures, the disease remains uncontrolled in Spain, and the herd prevalence in cattle increased from 1.52% in 2005 to 2.28% in 2018 [2,3].

An outbreak of TB in livestock may occur owing to the persistence of the mycobacteria within the herd (i.e., residual infection) or because of its introduction into a previously free herd. Epidemiological studies suggest that the most frequent cause in Spain is residual infection, but new infection as the result of sharing pastures, and other key resources with both infected herds and infected wildlife, is also a fairly important cause [4]. In the Iberian Peninsula, wild boar (*Sus scrofa*) and deer (mainly red deer, *Cervus elaphus,* but also fallow deer, *Dama dama*) are considered the main wild reservoirs hampering the eradication of TB, principally in the Mediterranean area [5]. However, recent studies suggest that European badgers (*Meles meles*) may be a potential reservoir of *M. bovis* infection in Atlantic Spain [6,7], especially since they are a recognized maintenance host in the United Kingdom (UK) and the Republic of Ireland (ROI) [8,9,10]. The badgers in both countries can maintain infection independently of other hosts and excrete mycobacteria in their sputum, urine, feces, and the pus from their wounds [11,12]. The results of a major field experiment performed in England (the Randomized Badger Culling Trial [RBCT]) have strongly implicated the badger in the transmission of *M. bovis* to cattle [13,14], although the transmission routes and dynamics are still unclear and the importance of badgers in regard to maintaining TB in cattle is currently controversial [15]. In this respect, medium- and long-term studies are valuable in regard to both understanding the epidemiology and temporal dynamics of TB and assessing the risk posed by wildlife species such as badgers [16,17]. This is especially relevant in areas that have become endemic (e.g., the hot-spots). Understanding the spatial ecology of badgers is also relevant in regard to disentangling key aspects of their role in TB epidemiology and providing a basis on which to control the transmission of TB [18].

Badger–cattle interactions are difficult to study in the field, owing to the elusive nature and nocturnal behavior of badgers. Studies carried out with tracked individuals have shown that, despite avoiding direct contact with cattle, badgers prefer cattle pasture, which is, therefore, the most probable means of indirect transmission [19,20]. Furthermore, badgers’ dispersion may not only facilitate the spread and maintenance of TB within badger populations but also the transmission to and from cattle [21,22]. In this respect, the shift in those areas in which badgers are active was related to the increase in the incidence of TB in cattle during the RBCT [14,23]. Super-rangers were also more frequent than expected in this species, which suppose a higher risk for the transmission of TB (and other diseases) [24,25].

In the UK and ROI, the landscape structure, wildlife communities, and farmland practices may modulate the badger’s spatial behavior and, therefore, its role in TB transmission and maintenance. Here, for the first time outside the British Isles, we document trends over six years in TB prevalence in cattle and badgers in a hot-spot in the Asturias region (Atlantic Spain) and investigate the ranging behavior and habitat use of badgers in relation to the paddocks used by cattle. Our aim was to assess whether badgers could be playing, or might play in the future, an important role in maintaining TB in this area. The study was carried out in the hot-spot of Parres, where the badger density is high, and there is evidence of strains of TB that are shared by cattle and badgers [7].

## 2. Results

### 2.1. Medium-Term Description (2012–2018) of TB in Both Cattle and Badgers in the Hot-Spot Area

#### 2.1.1. Cattle

The first focus of TB in cattle in Parres (the hot-spot area, see Figure 1) was detected on a beef farm in 2012 and had extended to nine herds by 2016. Of the forty-five cattle farms in the area, 28 (3 dairy and 25 beef) tested positive for TB when employing the official single intradermal tuberculin test (SITT) and culture technique between 2012 and 2018, and eight of them were subjected to “stamping out” (complete depopulation) and then restocked. The prevalence of TB in herds of cattle in this area was, by years, 2.22% (1/45) in 2012, 4.44% (2/45) in 2013, 15.55% (7/45) in 2014 and 2015, 20% (9/45) in 2016, and 4.44% (2/45) in 2017, while all tested negative to SITT in 2018 [3] (Figure 2). Significant differences were observed between 2014/2015 and 2018 (*p* = 0.0069), 2016 and 2017 (*p* = 0.0238), and 2016 and 2018 (*p* = 0.0016). The individual prevalence of TB in cattle in the area during the study period was less than 0.3% [3].

*Mycobacterium bovis* isolates were characterized by spoligotyping as SB0828 and the Mycobacterial Interspersed Repetitive Units—Variable Number Tandem Repeats (MIRU-VNTR) profile as 5-5-3-4-5-9-3-3-6 on all farms.

In the period in which collars were used on badgers (2013–2015), 14 farms tested TB positive, of which six were subjected to stamping out. On the remaining farms, only positive animals were slaughtered, while those that were negative were officially subjected to movement control (immobilization) for six months.

#### 2.1.2. Badgers

MTC species were isolated and identified by means of quantitative Polymerase Chain Reaction (qPCR) from 7/83 (8.43%) badgers. The isolates were identified as *M. bovis* and were characterized by employing spoligotyping as SB0828 and VNTR profile 5-5-3-4-5-9-3-3-6 (the same strain as in cattle). The proportion of TB in badgers was, by culture, 5.55% (2/36) from 2012 to 2015 (one badger was from 2012, and the other was from 2013) and 10.64% (5/47) from 2016 to 2018 (one badger was from 2016, and four were from 2018) (Figure 2). Positive badgers were spread over the whole area. No significant differences between years were observed in this period. Gross lesions were observed in six out of the seven badgers tested positive by culture (one positive badger from the first period did not have any gross lesions). Gross lesions in one of the TB-positive badgers in the first period (2012–2015) consisted of miliary areas of caseous necrosis and mineralization located in the bronchial and mediastinal lymph nodes (LNs) and lungs. Two of the TB-positive badgers in 2016–2018 had similar miliary lesions in submandibular or retropharyngeal LNs. In contrast, the three remaining TB-positive badgers from the second period had granulomatous lesions (from 2 mm to 1 cm) in the lungs and submandibular, retropharyngeal, mediastinal, bronchial, and hepatic LNs (Figure 3). One of these badgers also had lesions in the mesenteric LN (Figure 3).

Thirteen out of the 19 animals sampled in 2018 (68.42%) tested positive when using ELISA.

### 2.2. Spatial Ecology of Badgers

#### 2.2.1. Activity, Movement, and Home Range

One of the eleven animals tracked (ID 8) was not considered for spatial analyses owing to collar failures (see Material and Methods Section). All the monitored badgers tested serologically negative for TB. During the daily activity period, the badgers moved at an average speed of 0.28 km/h (maximum 5.12 km/h). The average travel speed varied among seasons (F_3,8941_ = 23.36, *p* < 0.001); there was also a significant effect of the interaction between hour and sex on the travel speed (F_13,8941_ = 5.44, *p* < 0.001; see Figure 4). The average day range (DR) and maximum range (Dmax) were 1.88 ± (SD) 1.20 km and 1.99 ± 0.71 km, respectively. One outlier (ID7) was excluded from the Dmax estimations and models; the badger in question behaved in a dispersive manner in winter (Dmax 18.28 km; see below). The models showed seasonal differences mediated by sex in both movement parameters (Season*Sex: F_3,1029_ = 8.73, *p* < 0.01, and F_3,11_ = 61.60, *p* = 0.01, for DR and Dmax, respectively), with lower rates during autumn and winter than during spring and summer, but only for the females (see Figure 5).

After excluding the outlier ID 7 in winter, the home range size varied significantly between sexes (Khr95: F_1,14_ = 4.84, *p* = 0.02; Kcr50: F_1,14_ = 4.52, *p* = 0.02) and among seasons (Khr95: F_3,14_ = 3.87, *p* = 0.01; Kcr50: F_3,14_ = 3.55, *p* = 0.008). Independently of the season, the males occupied larger areas than the females (Khr95: males 149.78 ± 25.84 ha and females 73.37 ± 22.91 ha; Kcr50: males 29.83 ± 5.69 ha and females 13.59 ± 5.00 ha). According to the Tuckey test, and similarly for Khr95 and Kcr50, the home ranges were smaller in autumn and—mainly—in winter than in summer, while the differences in home ranges were not significant between spring and summer (see Figure 6). The interaction between sex and season was not significant in neither the Khr95 nor the Kcr50 models and was, therefore, excluded from the final models.

#### 2.2.2. Badger Habitat Selection

The resource selection function showed the relevance of each type of land use in regard to explaining the badgers’ habitat selection. Badger habitat selection was not random, with a significant variation in land uses that varied with season (seasonal*land use: F_21,107363_ = 16.66, *p* < 0.01; see Figure 7). Apple orchards were preferred in all seasons other than winter, when badgers preferred pastures. Some land uses were consistently avoided in all seasons: shrubland, eucalyptus plantations, and urban areas.

With regard to patches on herds of cattle pasture, the model showed seasonal differences in the badgers’ preferences (TB status*season: F_3,26905_ = 70.71, *p* < 0.001), with badger use being higher on the paddocks used by TB-positive herds than on those used by negative herds in spring and summer, although this pattern was the opposite in autumn and winter.

Finally, badgers were frequently located near to farm buildings in the study area. Eighteen farms were visited by collared badgers. The mean number of locations on farms visited by badgers was 19.32 ± 30.87 (1–104). No significant relationship was found between the number of badger locations close to farm buildings and the cattle’s TB status (Z = –0.155, *p* = 0.877).

## 3. Discussion

We described the medium-term outcomes of TB infection in cattle and badgers in a hot-spot area of Atlantic Spain and studied, for the first time in continental Europe, the badger’s spatial ecology from an epidemiological perspective in order to describe risky behavior in regard to the transmission of TB to cattle. We found evidence of TB transmission between cattle and badgers and hazardous behavior by badgers.

The best available estimates for TB transmission originate from the UK and suggest that badger-to-cattle transmission causes between 1% and 25% of new outbreaks of TB, with the most likely value being 6% [26]. This implies that at least 75%, and possibly as many as 94%, of the TB-affected herds would be infected by other herds of cattle [26]. This situation might be similar in Spain, since 20% of new outbreaks of TB in cattle in 2006–2011 were associated with wildlife reservoirs (mainly wild boar and, to a lesser extent, red deer and fallow deer) [4], although that frequency is likely lower in Atlantic than in Mediterranean Spain [5]. However, given the potential role of badgers in explaining the spread and persistence of TB in cattle [27], the description of the spatial ecology of the species shown in our study provides a basis on which to understand the potential role of this species in TB epidemiology in Atlantic Spain (Figure 8). The herd prevalence of TB in cattle in Asturias significantly decreased from 0.19% (28/14,695) in 2012 to 0.08% (13/16,500) in 2018 (Fisher’s exact test; *p* < 0.05), although geographical “hot-spots” such as Parres remained, where the herd prevalence was as high as 20% in 2016 [3]. The number of TB-positive herds in Parres was significantly higher in 2014 compared to 2018, which suggests that TB prevalence decreased in this period. Our results indicate that, while TB in the cattle in Parres has decreased in the last few years, owing to the pressure of official national eradication campaigns, the TB in badgers has increased notably (from 5.55% to 10.64%), thus enhancing the importance of badger ecology that could potentially transmit TB back to cattle in the future (Figure 8). This conclusion was reached for four main reasons, the first of which is that the incidence of TB in badgers in the hot-spot area is much higher at present (21% and 68% by culture and serology, respectively, in 2018) than in the rest of Asturias. In this regard, we have to take into consideration that the TB in badgers in Asturias estimated by culture from 2008 to 2012 was 8.2% in a sampling of 171 road traffic accident (RTA) animals (two of the 14 positive badgers were from Parres) [7]; however, culture data indicate that TB has decreased in this species in Asturias from 2012 to <1% at present [28]. The second reason is that the VNTR analysis carried out in the present study demonstrated that the same *M. bovis* isolate was present in both cattle and badgers in Parres, which is indicative of previous interspecies transmission, presumably from cattle to badgers. The third reason is that the TB gross lesions found in badgers in recent years are indicative of a higher percentage of generalized lesions than those observed in previous studies in Asturias [6,7], affecting not only the head area and thoracic cavity but also the abdominal cavity (three out of five TB-infected badgers had lesions in hepatic LNs and one of them also in the mesenteric LN), thus increasing the potential for fecal excretion, environmental contamination and the risk of intra and interspecies transmission in the future [12]. Finally, badgers’ behavior and ecology indicate more activity on TB positive farms subjected to control movement than on negative ones, thus increasing the risk of indirect interspecies transmission. Nevertheless, as pointed out above, we must take into account that cattle movements to TB-endemic regions located in central Spain have been responsible for new outbreaks in the hot-spot area [28] and are suggested as a relevant risk factor. Additionally, TB-infected wild boar (estimated prevalence of 5%) sharing the same *M. bovis* strain were found in the area [28], likely contributing to a possible three-host maintenance community.

With regard to behavioral patterns, the badgers’ movements were characterized by a marked seasonality. The animals were more active, undergoing greater daily and dispersal displacements and occupying wider home ranges (females only) in spring and summer than in autumn and winter. This result is consistent with previous studies and can, to a great extent, be explained by the availability of food [29,30,31]. When food is available to a lesser extent—spring and summer—the animals need to explore larger territories. During the spring and summer, the beef herds graze in communal (shared) pastures in a mountain chain close to the study area (“Sierra del Sueve”). This means that, in those periods in which the risk of interaction between cattle and badgers is potentially higher, the cattle are at higher altitudes in areas in which badger abundance is lower [32]. The use of communal pastures in spring and summer could, therefore, be a means to reduce the risk of transmitting TB to badgers, although it could pose a new risk of infection associated with the aggregation of several herds [4]. However, in order to interpret the risk of TB transmission by indirect contact, it is necessary to keep in mind the wide time window (for weeks and even months) in which *M. bovis* survives in the environment and remains infective for potential host individuals (expected in more than 50 and 80 days in water and soil, respectively; see Fine et al. 2011 [33]).

Earthworms and fruit are considered staple resources for badgers in Northern Spain [34]. Both earthworms and fruit have a marked seasonality. In summer the lack of rain and high temperatures limit the activity of worms and, therefore, their availability for badgers [35]. In addition, fruit is restricted to summer and early autumn, when worm availability is minimal. Heterogeneous landscapes with pastures and fruit-plantations, therefore, provide food for badgers all around year. However, shrubland, pine and eucalyptus plantations, and urban areas were consistently avoided throughout the year, since they are poor in earthworms and do not provide adequate refuge for setts [34,36]. Our results suggest that the seasonal variation in habitat usage is driven by food rather than shelter [36,37,38] in areas like Parres where hedgerows are well preserved and the availability of refuge is not, therefore, critical. From an epidemiological point of view, these results suggest a higher risk of contact between cattle and badgers in those humanized areas in which a large diversity of land uses and, therefore, food resources, allow them to reach higher population densities. In this respect, some European policies concerning parceling of land could be reducing the risks of TB transmission, but their consequences in regard to biodiversity should also be considered before making large-scale recommendations [39].

Closely related to the previous result, we found that badgers preferred paddocks used by TB-positive herds in spring and summer. As stated previously, during the spring–summer period, the beef herds graze in communal pastures in the neighboring mountain area. However, during the time that the global positioning system (GPS) collars were used (2013–2015), 14 cattle farms were TB positive; a stamping out was carried out on six of them, while only positive animals were slaughtered on the others. In this situation, the negative animals were kept officially immobilized during the following six months, remaining on the plots close the farms’ buildings, and these plots consequently underwent a greater intensity of grazing (higher for TB positive herds than for those that were TB negative, which were in the communal pastures in the mountain). In Atlantic Spain, in territories with a mosaic of forests and pastures, an intense use of livestock favors the presence of high loads of earthworms (±47.73 individuals/m^2^) [40,41], thus making those particular pastures more attractive to badgers. In the UK, cattle pasture was also selected by badgers [19], even when it did not imply the existence of direct contact between badgers and cattle, but did imply a high potential for indirect contact with infected natural resources. Positive TB badgers are known to excrete *M. bovis* bacilli in their urine, feces, sputum, or bite-wound samples, thus leading to environmental contamination [12,42,43]. In our concrete study case, the risk would be even higher when dealing with pastures grazed by cattle from TB-positive herds.

During visits to farm building, badgers might also have access to cattle feed from feed-sheds, cattle sheds, silo yards, and cattle troughs, and then defecate and urinate directly onto that cattle feed. This constitutes a potential source of transmission of TB from badgers to cattle, or vice versa. However, no positive associations between badgers’ use of farm buildings and the infection status of either badgers or cattle were observed [19,44,45,46].

In addition, the badgers monitored displayed some particular types of behavior with potential implications for TB epidemiology that deserve further discussion. The activity level was quite constant throughout the activity period, suggesting that food is close to the badgers’ setts [30]. However, the activity period was shorter for females than for males (Figure 3) [30]. Assuming no sexual differences at feeding time, this result could be related to the time that males spent pursuing social activities, namely interacting with conspecifics, scent marking, or patrolling their territory, which was longer than that of females [47]. Social activities are more intense in high-density areas than in low-density areas [48], and this could be related to a higher potential risk of males rather than females becoming infected with TB and spreading the infection to cattle [8].

Our results suggest that most badger movements occurred within the boundaries of their social group´s territory. However, during our study, we recorded a movement by one male of 18.3 km, which could be considered as a super-ranging-dispersing badger [24,25]. Although it is still unclear why some male badgers maintain this behavior, it may be significant for the transmission and control of TB in badgers and in the wildlife–livestock interface, since super-rangers may act as relevant spreaders of TB infection.

An average Khr95 of 149.78 ± 25.84 ha for males and 73.37 ± 22.91 ha for females was obtained in our study area. The size of the badger’s home range may vary to a large extent throughout Europe, from only 10 to 50 ha in the pasture-dominated landscapes of the British Isles, and up to 2,440 ha in the Bialowieza, Poland [30]. This could, in a first instance, be related to the availability of food, with smaller areas being occupied as density and resources increase. In addition, differences in badgers’ ranging behavior have been related to TB status in the UK, with ranging behavior becoming increasingly abnormal as the disease progresses [49,50]. Our results showed seasonal differences for females (but not males), which occupied smaller areas in autumn and mainly winter than in spring and summer, that cannot be related to TB status since all the animals were TB negative (at least by serology). This ranging behavior could indicate that males’ and females’ use of space is affected by different factors. This is, in the case of males, access to females, whereas, in that of females, it is the availability of food [29,36].

In conclusion, these results suggest that the TB in Parres is being jointly maintained at least by cattle and badgers (pending further studies in wild boar) and that the incidence of the disease is increasing in badgers over the years. Our results also depict an area with a high environmental potential for maintaining high-density badger populations. Land uses and landscape structure, linked with some practices of livestock management, provide a scenario with a great potential for badger–cattle interactions. These results, together with the previous evidence regarding the role played by badgers in the spread and maintenance of TB in Atlantic Spain [6,7], lead us to recommend the implementation of an integrated (population and health status) badger monitoring and control program (e.g., vaccination) in at least those Atlantic “hot-spots” with high TB prevalence in cattle.

## 4. Materials and Methods

### 4.1. Ethical Statement

All methods were employed in accordance with the relevant guidelines and regulations. All experimental protocols were approved by the ethical committees from Servicio Regional de Investigación y Desarrollo Agroalimentario del Principado de Asturias (SERIDA) and from the Government of the Principality of Asturias. The license reference numbers were 010/07-01-2011, PROAE 20/2015, and PROAE 47/2018. All data from cattle were derived from the official testing schemes and did not require animal experiment ethics approval.

### 4.2. Study Area

The study was carried out in Parres, in the province of Asturias, which is located in Northwestern Spain (43°23′ N; 5°14′ W). Asturias is characterized by an Atlantic climate, with a temperature range from –4 to 8 °C in the coldest months and abundant precipitation throughout the year (1400–2100 mm per year) [51]. The study area includes a well-studied high-density badger population (6 individuals/km^2^; see Acevedo et al. 2014 [32] that extends over 17 km^2^ (see Figure 1). The land is low-lying at 140–360 m above sea level and is naturally fragmented, with approximately 60% of the area occupied by farmed grasslands, tillage, and apple orchards, while the rest is forested.

### 4.3. Medium-Term (2012–2018) Description of TB in Cattle and Badgers in Parres

#### 4.3.1. Cattle

Culture and SITT data were available for cattle culled as part of the National Program for the Eradication of TB [3]. There are forty-five cattle farms in the study area (14 dairy and 31 beef). The herds (dairy or beef) are fed by means of grazing, silage, and hay. A field survey was carried out in order to determine the paddocks and buildings used by each farm, and the information obtained was stored, using geographic information systems (GIS).

#### 4.3.2. Badgers

Eighty-three badgers (*n* = 83) from Parres were necropsied for postmortem examination from 2012 to 2018: 2012 (*n* = 10), 2013 (*n* = 15), 2014 (*n* = 9), 2015 (*n* = 2), 2016 (*n* = 21), 2017 (*n* = 7), and 2018 (*n* = 19). The location of the 83 badgers was determined by using the Universal Transverse Mercator (UTM) coordinate system. Of these animals, 29 were trapped badgers that were subsequently euthanized, and 54 were RTA badgers found by gamekeepers in the area. As we did not have a similar representative “*n*” for every year during the study period, the results are presented in two periods, from 2012 to 2015 (*n* = 36) and from 2016 to 2018 (*n* = 47).

The gross visible lesions found during necropsy were recorded. Serial sections (0.2 cm) were taken from the lungs and LNs of each badger for further macroscopic observation. Tissue samples from the lungs and retropharyngeal, submandibular, tracheobronchial, mediastinal, hepatic, and mesenteric LNs were taken for bacteriological and molecular studies. For the purposes of culture, a pool of the aforementioned tissues was frozen at –20 °C, for no longer than two weeks before processing. Bacteriological studies were performed as previously described [6]. Briefly, the mycobacteria growth indicator tube (MGIT) liquid-medium system, Löwenstein–Jensen solid media with sodium pyruvate and Coletsos solid media were used to isolate members of the MTC. Pools of tissues (2 g) from the lungs, mandibular, retropharyngeal, tracheobronchial, mediastinal, hepatic, and mesenteric LNs were used for that purpose. After decontaminating the samples, using the BBL MycoPrep Becton Dickinson kit (BD Diagnostic Systems, USA), a MGIT liquid medium was incubated at 37 °C, for at least 6 weeks, using the automated BACTEC MGIT 960 (BD Diagnostic Systems, New Jersey, USA). Solid media were incubated at 37 °C, for at least 10 weeks. A qPCR to identify MTC species was performed on culture isolates, using the MTC forward-primer 5’-TAGTGCATGCACCGAATTAGAACGT-3’, the MTC reverse-primer 5’-CGAGTAGGTCATGGCTCCTCC-3’, and the TaqMan probe YY/BHQ 5’-AATCGCGTCGCCGGGAGC-3’, which amplifies a 184 base pair fragment [52]. MTC isolates were characterized by means of DVR-spoligotyping following the hybridization of biotin-labeled qPCR products onto a spoligotyping membrane (VISAVET ‘homemade’ membrane). The results were recorded in SB code, followed by a field of four digits according to the *M. bovis* Spoligotype Database website [28]. In order to confirm the similarity between the isolates from cattle and badgers in the same area, MIRU-VNTR typing was performed according to the protocol previously described [53], using nine (ETR-A, ETR-B, ETR-D, ETR-E, MIRU26, QUB11a, QUB11b, QUB26, and QUB3232) VNTR markers.

Moreover, sera samples from badgers from 2018 (*n* = 19) were tested, using an indirect ELISA (P22 ELISA) based on the P22 protein complex, to detect antibodies against MTC. The ELISA was performed as described by Infantes-Lorenzo et al. (2019) [54].

### 4.4. Badger Monitoring

A total of 18 setts were found in the study area (1.12 setts/km^2^). Eleven badgers (ID1, ID2, ID3.1, ID3.2, ID4.1, ID4.2, ID5, ID6, ID7, ID8, and ID9) were captured in steel-mesh box-traps baited with peanuts. The trapped badgers were anaesthetized with ketamine hydrochloride (0.1 mL kg^−1^), medetomidine (Domitor^®^; 0.05 mL kg^−1^) and butorphanol (Torbugesic^®^ 0.1 mL kg^−1^) administered by means of intramuscular injection. Their location, sex, age (cub/adult), and weight were recorded. Blood samples were collected from the jugular vein, and serum samples were tested by using the Brock (TB) Stat-Pak test [55] and the P22 ELISA.

The eleven badgers (4 females and 7 males) belonged to 8 social groups evenly distributed in the study area (Figure 1 and Table 1). They were monitored between June 2013 and December 2015, using GPS–GSM collars (Microsensory System, Córdoba, Spain). Their collars were programmed to provide one location each 30 min, from 19:00 to 08:00 h (when badgers’ activity is expected to be higher) [30], and one location each two hours, during the rest of the day, in order to maximize battery life. Fix-rate success was 59.3%, and positional error was 25 m. Each GPS location registered an identification of each animal (ID), date, time (solar time), geographical coordinates, and location acquisition time (LAT). According to the last parameter, GPS locations with LAT > 255 s were removed because they were considered anomalous relocations. In addition, GPS locations for the day of collar deployment were also discarded in order to avoid the inclusion of capture-induced behaviors in the analyses. One of the eleven animals tracked (ID 8) was not considered for spatial analyses, owing to collar failures.

### 4.5. Data Analysis

#### 4.5.1. Basic Parameters of Badger Spatial Ecology: Activity, Movement, and Home Range

The straight-line distance between each of the consecutive fixes divided by the time that elapsed between them (i.e., speed; km/h) was used as a measure of the animals’ activity pattern [56]. Since the frequency of locations is not sufficiently high to estimate the real distances traveled [57], the estimations of speed were used solely as an activity index in order to characterize individuals’ activity patterns. DR (i.e., daily distance traveled by an individual) and Dmax (i.e., maximum distance between locations in a seasonal home range) were also estimated for each individual and season (winter, spring, summer, and autumn) in order to characterize individual movement patterns as a proxy of the capability to spread pathogens [24]. Here, “day” is considered as an activity period that extends from 19:00 h of one day to 08:00 h of the next. Distances and speed were calculated by using programmed functions in the R 3.3.1 in R Core Team 2016 program.

Kernel estimates were used to determine individual home ranges and their seasonal variation [58,59]. We estimated the Kernel 95% as a home range estimate (Khr95) and Kernel 50% for the core range (Kcr50), using the ‘adehabitat’ R package [60]. We calculated the bandwidth or smoothing parameter ‘h’ by using the reference method [61], because the least-squares cross-validation method failed to converge for some of the animals with large sample sizes [62]. The minimum number of relocations per individual employed to estimate seasonal home ranges was 25 according to previous studies [61,63].

General lineal mixed models, including the individual as a random effect factor, were used to assess differences between sexes and among seasons in the spatial ecology descriptors. The R package ‘effects’ were used to plot the relationships between the dependent and independent factors (including interactions) in the models [64]. A protocol for data exploration was applied by following the recommendations of Zuur et al. (2010) [65], and assumptions were checked on the residuals of the model.

#### 4.5.2. Badger Habitat Selection

We assessed habitat selection and the way it varied between seasons by employing within-home-range resource-selection functions [66]. The availability of resources was sampled by using randomly generated points within each Khr95 individual seasonal home range. The number of random points was, in each case, ten times that of the locations (see Table 1). This approach allowed us to identify positive/negative selection as those land uses with a probability of use higher than 0.1, which is the probability expected by chance. We compared “used” versus “available” by using logistic regression models [67] in which the individual was included as a random-effect factor, and land uses and season, and their interaction, as fixed factors. Land uses were obtained from the thematic regional cartography, with a scale of 1:5000 [68]. Eight different land uses were considered in line with previous studies [34]: woodland, shrubland, pastures, cultures, apple orchards, eucalyptus plantations, pine plantations, and urban areas (see Figure 1). The information regarding apple orchards was updated by using regional cadastral information [69]. The models were carried out in R with the ‘lme4’ package [70].

In a separate analysis, we characterized the relationship between GPS-collared habitat use and cattle farms. First, a hierarchical logistic regression was again used to determine the badgers’ seasonal preference for those paddocks pastured by TB-positive herds of cattle in relation to the use of paddocks by TB-free herds. In this model, the season and the interaction between positive/negative paddocks and season were included as fixed terms, and the individual was included as a random-effect factor. Finally, the use of farm buildings by badgers was also explored. For this purpose, and according to the positional error of the GPS, a buffer of a 25 m radius around each building was estimated and overlapped with badger relocations to determine the frequency of locations close to farms, along with their seasonality.

#### 4.5.3. TB Prevalence in Cattle Herds and Badgers

Statistically significant differences were evaluated by using Fisher’s exact test. The statistical test was carried out by using SPSS Statistics 25 (IBM, New York, USA) and interpreted considering a *p*-value of 0.05 as being indicative of statistical significance.

## Figures and Tables

**Figure 1 pathogens-08-00292-f001:**
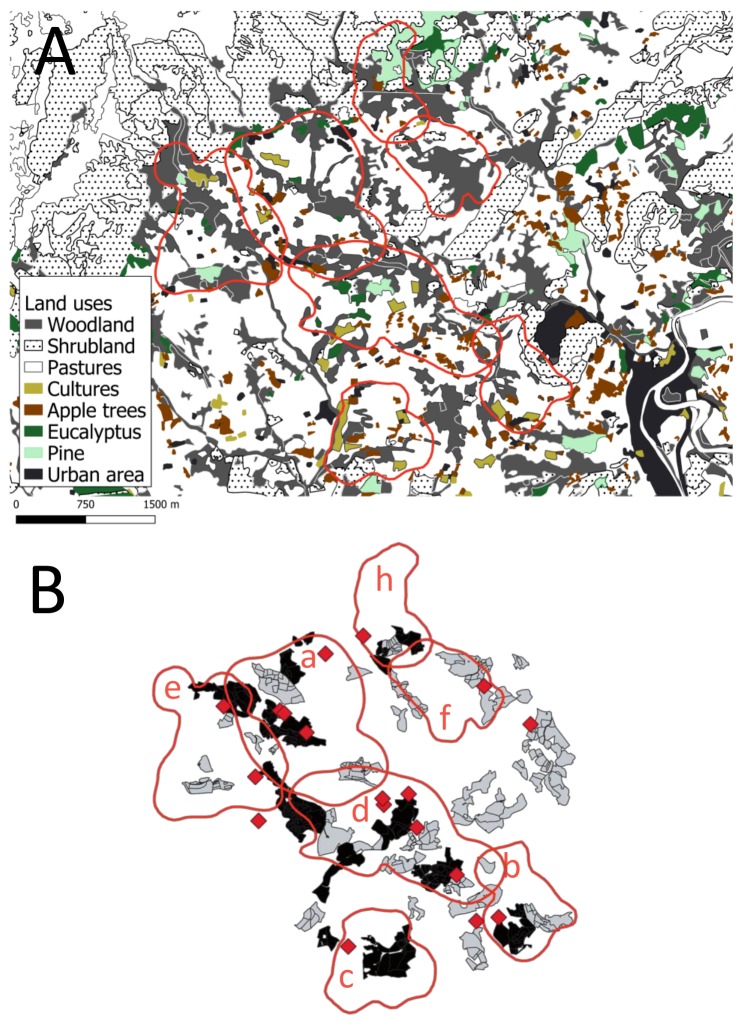
Study area (the hot-spot, namely Parres) in Asturias (Atlantic Spain), showing (**A**) landscape structure, including the land uses considered in badger (*Meles meles*) habitat selection analyses; and (**B**) an example of badgers’ seasonal Kernel 95% home ranges by social group (a–h; red polygons), location of badger setts (red diamonds), and paddocks used by the 14 tuberculosis positive (black paddocks) and 31 negative (grey paddocks) herds of cattle in the 2013–2015 period.

**Figure 2 pathogens-08-00292-f002:**
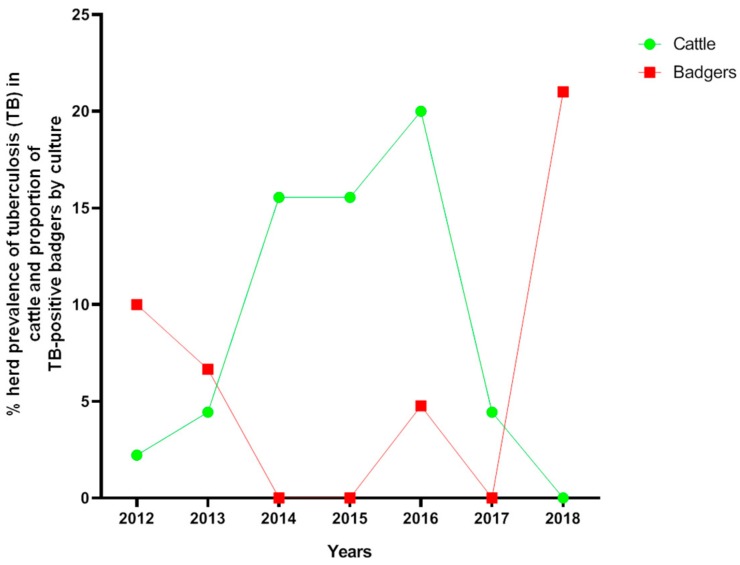
Trends in the herd prevalence of tuberculosis (TB) in cattle and the proportion of TB-positive badgers by culture in Parres, a hot-spot area of Atlantic Spain.

**Figure 3 pathogens-08-00292-f003:**
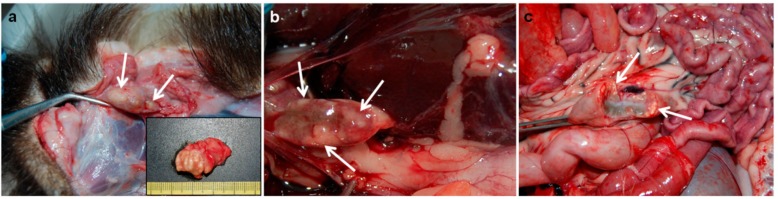
Gross lesions observed in a tuberculosis-positive badger obtained by means of culture from the hot-spot area in 2018. Submandibular (**a**), hepatic (**b**), and mesenteric (**c**) lymph nodes show tuberculous lesions consisted of areas of caseous necrosis and mineralization (arrows).

**Figure 4 pathogens-08-00292-f004:**
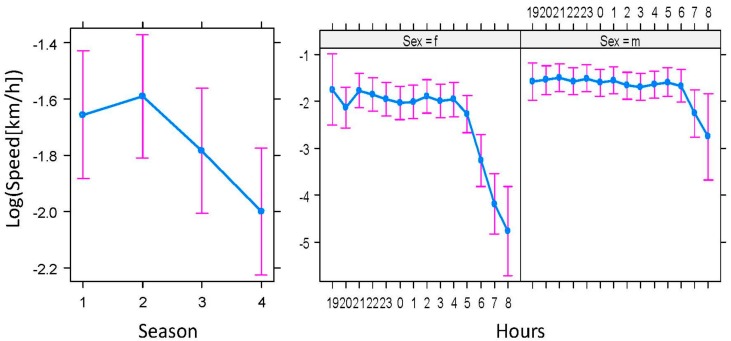
Differences in the activity of badgers (*Meles meles*) in Atlantic Spain: season (1—spring, 2—summer, 3—autumn, and 4—winter) and interaction between sexes (f—female and m—male) and hours during the daily activity period (from 19:00 to 08:00 h).

**Figure 5 pathogens-08-00292-f005:**
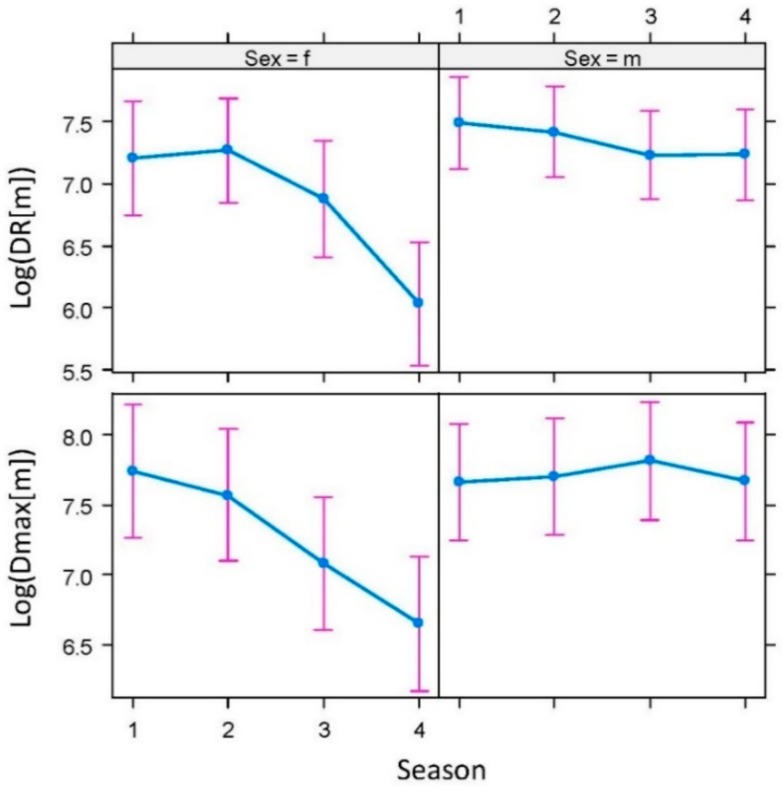
Sex-biased (f—female and m—male) seasonal (1—spring, 2—summer, 3—autumn, and 4—winter) differences in the movement rates (DR—daily distance traveled by an individual and Dmax—maximum distance between locations in a seasonal home range) of badgers (*Meles meles*) in Atlantic Spain.

**Figure 6 pathogens-08-00292-f006:**
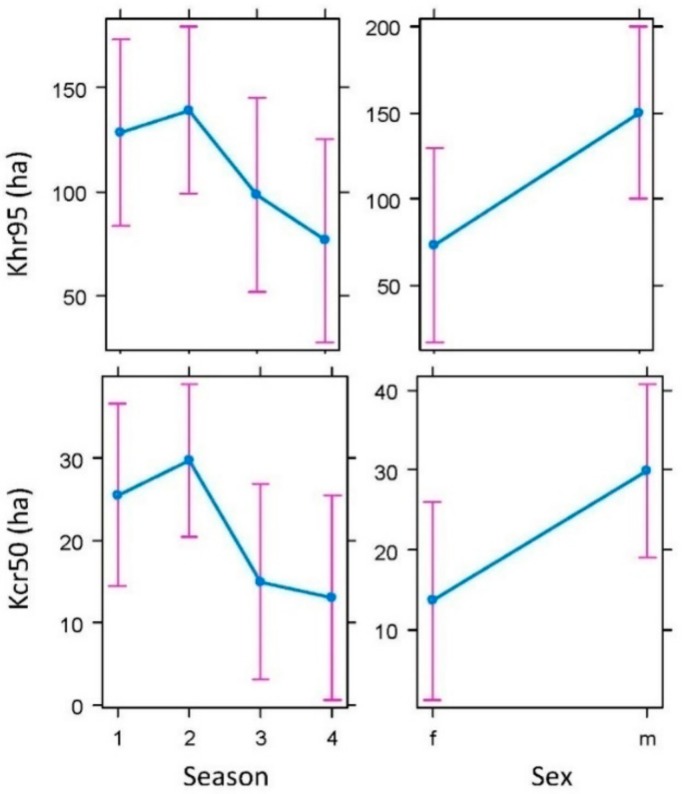
Differences in regard to seasons (1—spring, 2—summer, 3—autumn, and 4—winter) and sex (f—females and m—males) in home range size of badgers (*Meles meles*) in Atlantic Spain, both kernel 95% (Khr95) home range and kernel 50% (Kcr50) core area (see text for details).

**Figure 7 pathogens-08-00292-f007:**
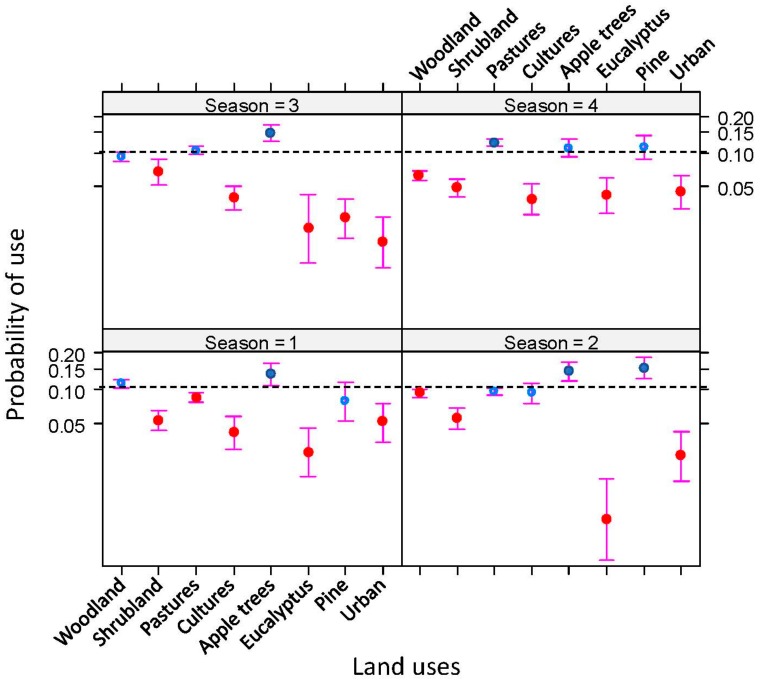
Seasonal (1—spring, 2—summer, 3—autumn, and 4—winter) differences in badgers’ (*Meles meles*) habitat selection in Atlantic Spain (see text for details). Dashed line shows the probability of occurrence expected by chance (sample prevalence). Over and under the dashed line are those land uses positively selected (blue) and avoided (red solid), respectively. Open symbols represent those land uses that are used according to their availability in the study area.

**Figure 8 pathogens-08-00292-f008:**
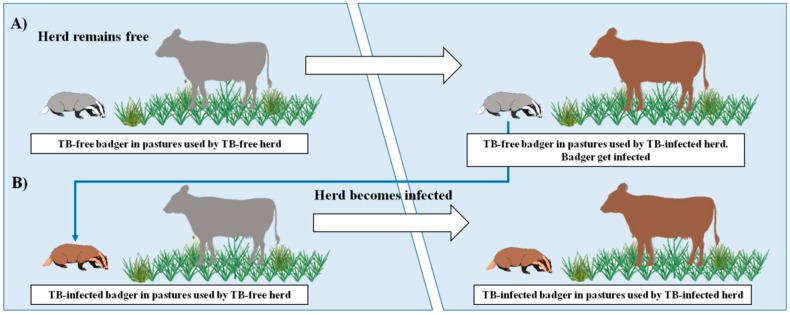
Hypothetical scenario in a hot-spot area. (**A**) In our hypothesis, a TB-free badger feeds in pastures used by a TB-infected herd and the badger subsequently becomes infected, likely by indirect contact owing to environmental contamination. (**B**) This badger may later use TB-free pastures and excrete mycobacteria into the environment. When a TB-free herd goes to the former pasture, it is indirectly infected. This model could allow the transmission of the mycobacteria between TB-infected and TB-free herds, linked by badgers. Gray shading: not-infected. Brown shading: infected.

**Table 1 pathogens-08-00292-t001:** Basic parameters of badger (*Meles meles*) monitoring and spatial ecology in the study area and their seasonal variation. Sex (M: male or F: female), social group (a to h), number of relocations, number of nights with data, and the home range (using kernel methods) are provided for each individual and season. “Total” summarizes all the information obtained in this study.

	Individual ID										Total
	1	2	3.1	3.2	4.1	4.2	5	6	7	9	
**Sex**	M	F	M	F	M	F	M	F	M	M	6 M & 4 F
**Social Group**	a	b	c		d		e	f	g	h	8
**N Relocations**											
Spring	144		359		409			1074			1986
Summer	312	58	209	727		164		616		254	2340
Autumn	377			1209						1082	2668
Winter				444			956		742	725	1911
Total	833	58	568	2380	409	164	956	1690	742	2061	8905
**N Nights**											
Spring	19		26		42			82			169
Summer	63	9	17	55		16		44		20	224
Autumn	59			90						84	233
Winter				45			83		72	48	248
Total	141	9	43	190	42	16	83	126	72	152	874
**Home Range (ha)** **Kernel 95%**											
Spring	212		93		203			91			NA
Summer	193	80	154	109		102		91		72	NA
Autumn	170			59						38	NA
Winter				22			168		3358	26	NA
Total	175	80	120	74	203	102	168	83	3558	42	NA
**Home Range (ha)** **Kernel 50%**											
Spring	59		25		31			17			NA
Summer	51	18	45	20		16		26		17	NA
Autumn	37			9						6	NA
Winter				5			16		225	7	NA
Total	44	18	31	10	30	16	16	20	225	7	NA
**Daily Distance (m)**											
Spring	1193		2305		2861			2328			2329
Summer	1198	803	2531	2277		1664		2359		1232	1831
Autumn	1256			1630						1382	1449
Winter				814			2618		2212	1271	1907
**Dispersal Distance (m)**											
Spring	1969		1637		3781			1924			NA
Summer	1961	1322	1799	3339		1986		1620		1329	NA
Autumn	2216			2040						1528	NA
Winter				1324			2793		18286	1300	NA

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
