# Peer review of "Tuberculosis Epidemiology and Badger (Meles meles) Spatial Ecology in a Hot-Spot Area in Atlantic Spain"

_pathogens, 2019, doi:10.3390/pathogens8040292_

Round 1
Reviewer 1 Report
Overall, the paper presents interesting and important data indicating a high likelihood that badgers are somehow contributing to the maintenance of TB in northern Spain. Although a largely descriptive study, the findings (particularly the upsurge in TB in badgers when TB in cattle had fallen to zero do strongly (but not conclusively) implicate badgers in TB maintenance.
The paper is well written, but would benefit from input from a native speaker/writer of English.... quite a few of the comments below are suggested wording changes/edits aimed at improving the English.
Specific comments/questions (according to line number) are:
L25: ‘… hazardous behaviour in relation to TB transmission between cattle and badgers’
L28: Use ‘areas’ rather than ‘patches’
L30: ‘Ten badgers….
L32: ‘….increased from 1.8% in 2012 to 16.4% in 2016 then decline to 0.0% in 2018.’
L35-37: Range refers to an area, not a distance. Reword: ‘The average of the index of the daily distance travelled…..’. ‘One badger dispersed 18.3km.’ It might be better to present home range size (K95) here, rather than distances travelled?
L50 ‘…cattle increased from 1.52% in 2005 to 2.28% in 2018.’
L54 ‘.. but new infection…’
L56 ‘.. important cause’
L60 ‘…, especially since it is a recognised maintenance host in…’
L66 ‘… medium-term?’
L75 ‘.. displace and disperse’ means what?
L80… odd English
L82-89: A clearer statement of the objective(s) of the paper would be useful here: Maybe something like: ‘Here, we document trends over 6 years in TB prevalence in cattle and badgers in a TB hot spot in the Astruias region, and investigate the ranging behaviour and habitat use of badgers in relation to there areas used by cattle. Our aim was to help assess whether badgers could be playing an important role in maintaining TB in this area.”
L94-97. The number didn’t initially appear to make sense; 1 herd in 2012 increasing to nine herds by 2016, but then 28 in total positive for TB by test and culture? But then I realized the 28 was for the whole period. Perhaps report as herd prevalences: 1/45 (1.63%) in 2012, 2/45 (3.6%) in 2013, etc… Were the 28 herds all different? Or did some herds become infected twice or more? Judged from Fig 2B, it looks like less than half of the cattle area was infected, suggesting that some herds were infected in multiple years?
L97. What does ‘stamping out’ mean? Complete depopulation? Were the ‘stamped out herds restocked? Or did the number of herds decline?
L104: ‘.. were subject to movement control…’ might be better than ‘immobilised’
L105: Fig. 1 is visually difficult with a great amount of detail masking it difficult to see the overall pattern, and the green home-range boundaries in Fig1A lack enough contrast to be readily distinguished.
L111: Fig 2. Perhaps refer to ‘Trends in the herd prevalence of TB in cattle and the individual prevalence of TB in badgers… rather than ‘Temporal progression…’ . The y-axis should be ‘TB Prevalence’ and should show the units of measure. The figure shows spill-over from cattle to badgers which implies cattle are the maintenance host, contrary to L321. It also shows spillback from badgers to cattle in 2018 which is also contrary to L231, but also is not supported by the data as there was no TB in cattle that year.
L120: Are the locations of 83 badgers known? In particular, were the four in 2018 clustered in one area? Or were the positives spread over the whole area.
L124: Why is this particularly interesting? What does it mean, if anything?
L129: Again, so what does this mean, if anything?
L137: There is no mention of collar failure in the Methods section
L162-170: Perhaps try to find a way to describe the effect first and the statistical support second: e.g.; Males occupied larger areas than females [stats] and ranges were largest in summer and smallest in winter (Fig 5; [stats])
L181: Wording: … were preferred in all seasons other than in winter, when badgers preferred pastures.’
L192: Perhaps refer to paddocks rather than plots?
L199: Medium term (six years isn’t long-term for a slow disease like TB).
L204: ‘…..TB transmission between cattle and badgers….’
L207-209: This is misleading because the main wildlife host in most of Spain is probably the wild boar, not badgers.
L212: Please give the denominator for Asturias herd prevalence – how many herds in the region? Is the downward trend statistically significant?
L214: There are no statistics in the results section showing a simple decrease in herd prevalence … there is obviously no consistent downward trend (linear regression, R2 = 0.0005)) … an inverted parabola (second order polynomial) provides a good fit (R2 = 0.79) , perhaps (for example) suggesting initially uncontrolled spread in cattle within Parres that was then brought under control by testing, stamping out, and movement controls.
L216: Again, there are no results showing a significant trend upward in badgers – the difference between 2012-15 (2/36) and 2016-18 (5/47) has no statistical supported (Fisher’s exact test, p = 0.45). It would also be interesting/important to know whether all of the TB+ves in 2018 were likely to be from the same social group/location in 2018 and from different groups in previous years…
L225: Wording: ‘…demonstrated that the same M. bovis isolate was present in both badgers…’
L227: Higher percentage than what? Presumably higher than other badgers elsewhere? If so, give a reference.
L229: Do hepatic lesions increases the potential for fecal excretion?
L231-233: Badgers preferred TB farms only in summer but avoided them in winter, so unless transmission only occurs in summer, the risk should balance out?
L256: Repeats results
L259: In summer, the lack of rain and high temps…
L264: ‘…suggest that the seasonal variation in habitat usage is drive by food rather than shelter [in areas like Parres?] where hedgerows are well preserved.’
L269: What is reparcelling?
L304: ‘.. a seasonal movement..’ suggests the badger went somewhere then came back in a different season? However, IDu7 was only monitored in winter, so it can’t be classed as a seasonal movement. Maybe this was simply a dispersal event where an animal left its home range and went somewhere else permanently (which animals, especially males, often do)?
L309-319: This paragraph has some ‘natural history’ interest, but the linkages to Tb status or infection risk is weak.
L321: ‘Being maintained’ suggests that badgers are being classed as true maintenance hosts of TB at Parres, but the descriptive evidence provided does not provide strong support for that conclusion. The evidence provided indicates that TB is being jointly maintained by cattle and badgers. The 2018 upsurge in badgers suggests that badgers were becoming infected in the apparent absence on Tb in cattle, but unfortunately there is no evidence that those badgers did not become infected several years previously, as they can carry infected for many years.
L345: Is this an isolated population? Or is the 17 km2 area just the area that has been studied?
L352: What is crow ensile? Silage?
L353: ‘determine the patches..” = ‘determine the area and building used by each farm?
L402: Excessive precision .. 59.3%
L406: Wording: ‘capture-induced’, rather than ‘conditioned’
L409: Not a proper sentence.
L410: The straight-line distance…
L434: Clumsy wording: ‘We assessed habitat selection and the way it varied between seasons by…’
L443, 444: Use ‘apple orchards’ instead of ‘apple trees’
L448,449: Use ‘use’ rather than ‘preference’.
L453: Why 50m, when positional error was 25m?
Reviewer 2 Report
The subject of this paper is of wide interest (wildlife-livestock disease interface) and TB is an emerging problem in some parts of Europe so it is a timely description of the situation in Northern Spain. The paper is essentially a descriptive epidemiological study with some ecological/behavioural data on the wildlife host added in. Unfortunately, I found it very difficult to follow in places (possibly because of poor English) and there was some erroneous use of epidemiological terms (e.g. incidence used when prevalence was more appropriate).
Clearly a great deal of effort went into the collection of the ecological data but a sample size of 11 animals over 3 years will limit what can be concluded and I found the results from this part of the study unconvincing (eg. how was annual variation controlled in analyses of the effects of season and sex on movement behaviour?).
My main concern was that the authors over-interpreted some of the findings and the main conclusions could not be supported by the data as follows,
The authors say their results suggest maintenance of infection in badgers, but this is not demonstrated and their results could equally arise because of periodic spillover from cattle. The authors say the 'incidence' (they mean prevalence) of infection in badgers is increasing, but they cannot conclude this from a change of 2/36 to 5/47 positives (there is considerable overlap of confidence intervals and a quick Chi squared test indicates no significant difference). The authors say there has likely been 'a high level of environmental contamination in the area' but no compelling evidence is presented to support this.The final closing conclusion that intervention in the badger population is required because TB is not being controlled in cattle by the national eradication programme seems perverse given that all herds were negative in 2018!
Reviewer 3 Report
The paper is well presented.
Author Response
Dear Editor,
We would hereby like to submit a revised version of our manuscript on “Tuberculosis Epidemiology and Badger (Meles meles) Spatial Ecology in a Hot-Spot Area in Atlantic Spain” by Acevedo et al. for publication as a “Research article” in Pathogens.
We greatly appreciated the useful suggestions of Editor which helped us to improve the quality of the manuscript. Please find a detailed response to the comments below. We have also provided a new version of the manuscript using the track changes mode.
On behalf of all co-authors,
Kind regards,
Dr. Ana Balseiro
1) Please indicate cattle individual prevalence along with herd prevalence, as this figure is better comparable to badger individual prevalence. We have included that information (lines 104-105).
2) Could you please, perhaps in the discussion, insert a small statement on cattle movements to TB-endemic regions, and state if these may have changed recently? This does not invalidate the study, but points to an additional relevant risk factor. We have included that statement (lines 249-251).
3) While I note the focus of this study is on badgers, I would like it if you could include a short paragraph in the discussion describing the mean infection prevalence in wild boar in the study area, with a comment on its possible contribution to a (probably) three-host maintenance community. We have included that statement (lines 252-254).
L279 driven. Done.
